# ROYAL SOCIETY
# OPEN SCIENCE

materials science/energy/chemical engineering

nanoporous structure, superhydrophilic, high transmittance, anti-fog property, glass

**Authors for correspondence:**
Lei Wang
e-mail: phy_wangl@zju.edu.cn
Deren Yang
e-mail: mseyang@zju.edu.cn

This article has been edited by the Royal Society of Chemistry, including the commissioning, peer review process and editorial aspects up to the point of acceptance.

# The preparation and characterization of uniform nanoporous structure on glass

Lei Wang[1], Likai Li[1], Youbo Liu[1], Shuxian Wang[1], Hui Cai[1], Hao Jin[2], Qingwen Tang[3], Wei Sun[3] and Deren Yang[1]

[1]State Key Laboratory of Silicon Materials and School of Materials Science and Engineering, Zhejiang University, Hangzhou 310027, People's Republic of China
[2]Zhejiang JinkoSolar Co., Ltd., Jiaxing 314416, People's Republic of China
[3]Bengbu Institute of Product Quality Supervision and Inspection Research, Bengbu 233000, People's Republic of China

LL, 0000-0002-7787-502X

A novel fabrication method of uniform porous structures on the glass surface is proposed. The hydrofluoric acid fog formed by air-jet atomization etches the glass surface to fabricate nanoporous structure (NPS) on glass surface. This NPS shows the enhanced average light transmittance of approximately 92.9% and the superhydrophilic property with a contact angle less than 1° which presents an excellent anti-fog property. Passivated by fluorosilane, the NPS shows nearly the superhydrophobic property with a contact angle of 141.2°. This fabrication method has shown promising application prospects due to its simplicity, low cost and efficiency, which can be easily applied to large-scale industrial production.

## 1. Introduction

The transparent porous glass has received increasing attention for its excellent properties such as high light transmission rate, low glare, large specific surface area and hydrophilicity or hydrophobicity. Superhydrophilic or superhydrophobic glass with high transmittance can be used in the fields of optics, architecture, energy, etc. [1–4]. The superhydrophilic glass has the spreading performance of water droplets, which can prevent the diffuse reflection of water droplets and reduce the static electricity on the surface, which can be used for anti-fog and suppression of small particles adsorption. For example, they can be used in myopia glasses, endoscopes, car windshields and other occasions. The

superhydrophobic glass can make the water droplets slide down quickly, inhibit the adsorption of large particles and sewage on the surface, and can be used in automobile rearview mirrors, solar cell module cover plates and other occasions. Generally, superhydrophilic and superhydrophobic glass can achieve self-cleaning of the glass surface and maintain high light transmittance [5]. Up to now, many methods, such as sol–gel synthesis [6,7], electron-beam deposition [8], self-assembly process [9] and chemical etching [10–14], have been reported to fabricate nanoporous structure (NPS) on the glass. However, most applications of porous glass need good rub resistance, which cannot be realized by the above methods solely. Thermal treatment or binder is one of the means to enhance the adhesion of NPS. For example, the industrial sol–gel method to produce anti-reflection glass consists of the growth of a layer of silica collosol, the adding of adhesion assistant and the thermal treatment [7]. Therefore, to develop a cheap and facile technique to manufacture NPS on glass is still a challenge. In our previous work, by chemical etching in alkali (NaOH) solution, a kind of nanoflake structure on glass has been reported. However, this structure showed the weak rubbing performance due to its large area and thin thickness of the structure.

In this work, a novel fabrication method for NPS is proposed. In the experiment, a hydrofluoric acid fog (formed by air-jet atomization) etching process is used. This novel fabrication method does not require the use of any chemicals other than hydrofluoric acid and can be achieved with a simple device. A sample can achieve high transmittance and superhydrophilicity in a short time. Passivated by fluorosilane, the NPS shows nearly the superhydrophobic property. So, the method is simple, cheap and high-efficiency, which makes it very convenient to produce on a large scale. The high light transmission and the excellent wettability make it possible to be an ideal outdoors application product, especially at the severe smog surroundings. For instance, normal glass curtain walls and solar cell module glass can be treated to form NPS layer before installation.

## 2. Material and methods

Commercial slide glass (Sail Brand, Cat. No. 7101, size: $25.4 \times 76.2 \times 1.1$ mm$^3$) was used as original samples. The samples were cleaned with acetone (CH$_3$COCH$_3$, analytical reagent (AR)), ethanol (C$_2$H$_5$OH, AR) and deionized (DI) water under ultrasonication in sequence and dried by compressed air before etching process. The etching process was conducted in a tubular Teflon container surrounded by the heating jacket to obtain different etching temperatures (303, 323, 328, 333 K). Figure 1 shows the schematic of the experiment set-up. First, the sample was placed in a tubular Teflon container, heated to a certain temperature for 5 min. At the same temperature, the hydrofluoric acid (HF, AR) fog formed by air-jet atomization (Yu Yue 403T, Yuwell-Jiangsu Yuyue Medical Equipment & Supply Co., Ltd) was adopted into the container to etch the samples at different time. After the etching process, the samples were taken out and rinsed with DI water and dried by compressed nitrogen. Surface modification was carried out in an enclosed Teflon container together with droplets of 1H, 1H, 2H, 2H-perfluorodecyltriethoxysilane and the container was set in an oven at 393 K for 120 min. The lid of the Teflon container was opened and the temperature was maintained at 423 K for 90 min to remove excess fluorosilane.

The morphology and structure of samples were observed by field emission scanning electron microscope (FESEM, Hitachi S4800) and atomic force microscope (AFM, Bruker Dimension Edge). The optical property of samples was measured by UV–VIS–NIR spectrophotometer (Hitachi, U-4100). The wetting property of samples was investigated by water contact angle measurement (Dataphysics, DCA20). Superhydrophobic glass surface composition was researched by X-ray photoelectron spectroscopy (Kratos, AXIS Supra).

## 3. Results and discussion

The morphology and structure of the porous layer on samples can be changed with the key parameters such as time, temperatures and hydrofluoric acid concentrations, etc. As an example, figure 2 shows the magnification SEM images of NPS after etching for different etching time (0, 20, 50, 60 s) where the etching temperature was fixed at 323 K and 9.6 wt% hydrofluoric acid solution was used. It can be found that the NPS on glass was formed, as shown in figure 2b–d. After 20 s etching, tiny pores began to form randomly on the surface (figure 2b). With the increase of etching time, the number and diameter of the pores increased continuously. After 50 s etching, the sample showed the uniform porous structure with the approximately 89 nm height and approximately 50 nm diameter

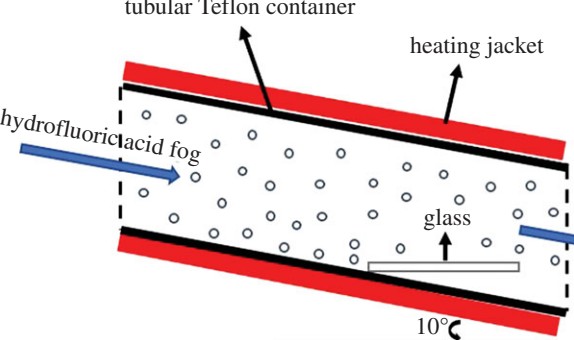

**Figure 1.** Schematic of the experiment set-up.

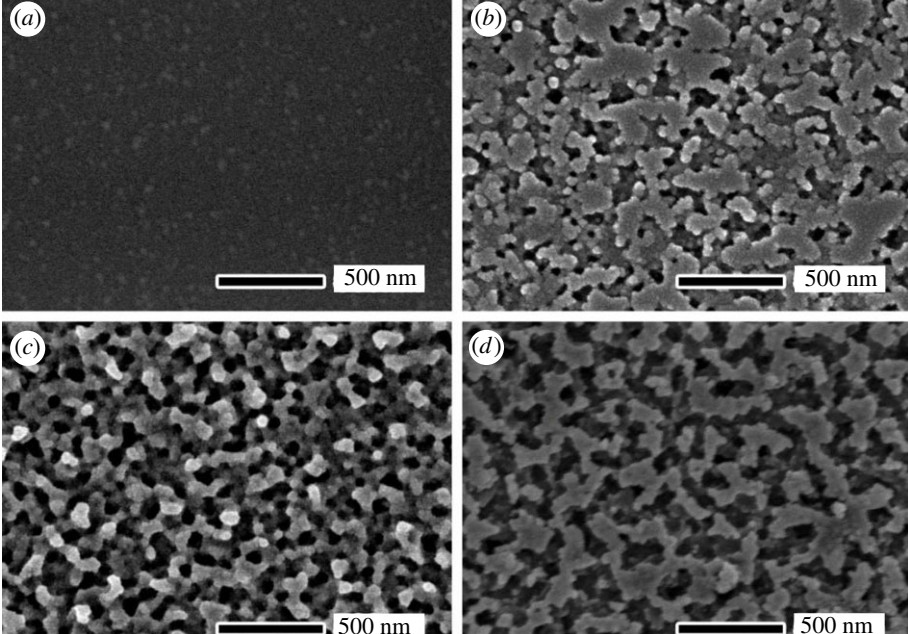

**Figure 2.** SEM images of the glass surface after etching for different time: (*a*) 0 s, (*b*) 20 s, (*c*) 50 s and (*d*) 60 s. The etching temperature was fixed at 323 K and 9.6 wt% hydrofluoric acid solution was used.

(figures 2*c* and 3*b*). With the etching time of more than 60 s, the pores connected to each other to form large channels whose sizes were larger than 100 nm.

The etching mechanism can be explained as follows. At the early stage, the micrometre-sized hydrofluoric acid droplets reached the sample surface and random tiny etching pores began to form. With the development of the etching process, the acid droplet was becoming smaller and smaller due to the etch reaction consumption and acid volatilization. Then, pores formed gradually and presented a large upper diameter and a small underneath diameter (figure 3*b*). This pore structure was beneficial to obtain a good anti-reflection effect [15]. When the etching time was extended further, more acid droplets reached the sample surface, and the diameters of pores were more and more enlarged to form the channels (figure 2*d*).

The uniform porous layer, whose pore diameter (e.g. 50 nm in figure 2*c*) is much smaller than the visible wavelength, can enhance the visible light transmission property [16]. The NPS introduces a refractive index gradient between air and glass, resulting in the decrease in the optical reflection. In other words, the transmittance of glass increases. Figure 4 shows the transmittance spectra of the un-etched sample and the samples etched for the different time at the visible light wavelength range. The un-etched sample showed an average transmittance ($T_{ave}$) of 90.9% in the visible spectra (400–800 nm). Then the $T_{ave}$ increased with the etching time going on. After 50 s etching, the highest $T_{ave}$ of 92.9% was obtained (figure 4). After 60 s etching, the $T_{ave}$ dramatically decreased to 88.9% because the porous layers were excessively etched to form the channel structures (figure 2*d*). The reason for this

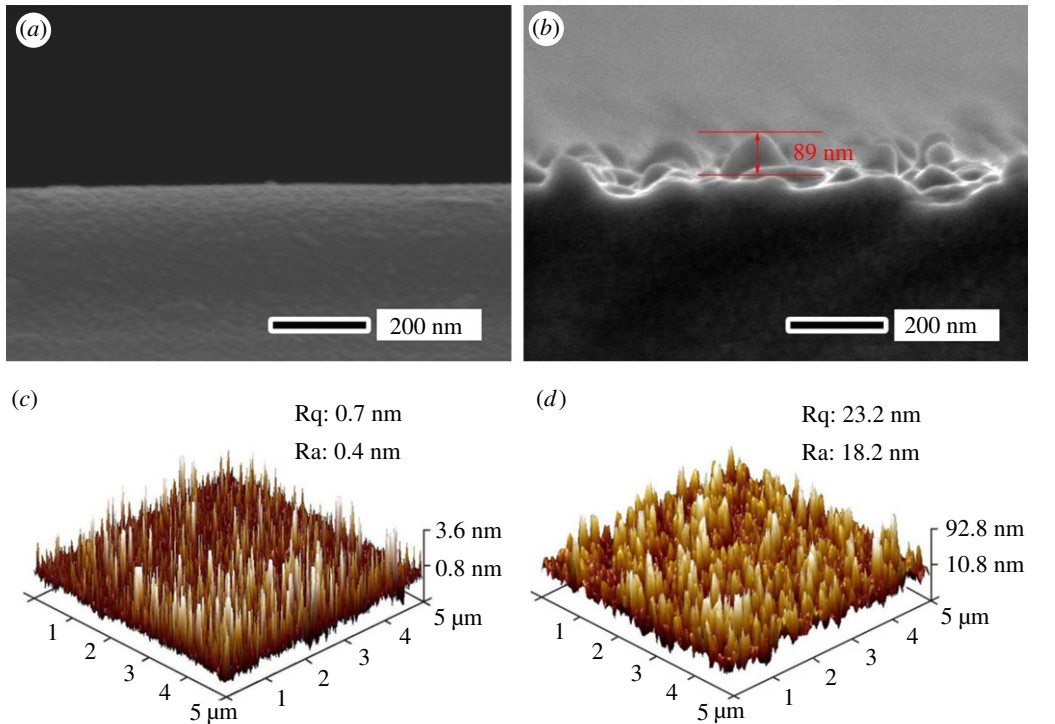

**Figure 3.** Cross-section SEM images and AFM images for the original sample (a,c) and the 50 s etching sample (b,d). Ra, arithmetic mean roughness; Rq, root mean square roughness.

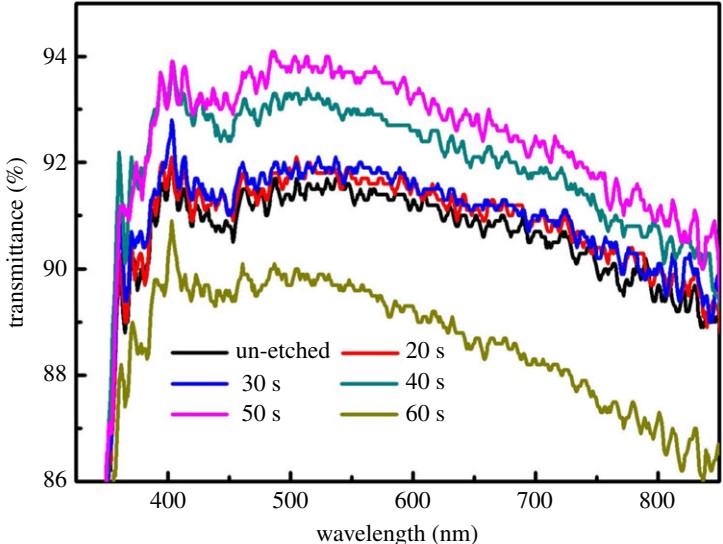

**Figure 4.** Transmittance spectra of the un-etched and the etched samples for different etching time.

phenomenon can be explained in this way. In the early stage of the etching reaction (20–30 s), the sample was gradually etched, and the surface began to produce tiny pores. As the reaction progressed (40–50 s), the distribution of pores became uniform and dense. According to effective medium theory [17]

$$n_1 = [n_g^2 f + n_{air}^2 (1 - f)]^{1/2},$$

where $n_1$, $n_g$ and $n_{air}$ are the refractive index of NPS, glass and air, respectively. $f$ is the fill factor, which is the percentage of NPS.

According to the analysis of the incident path of visible light, at the outermost surface of the sample, $f = 0$, $n_1 = n_{air}$. Down along the porous structure, the structure narrowed first and then widened, $f$ increased gradually from 0 to 1, $n_1$ increased from $n_{air}$ to $n_g$. It can be considered that NPS as the anti-reflection coating had a graded refractive index, so that transmittance can enhance. However, with the

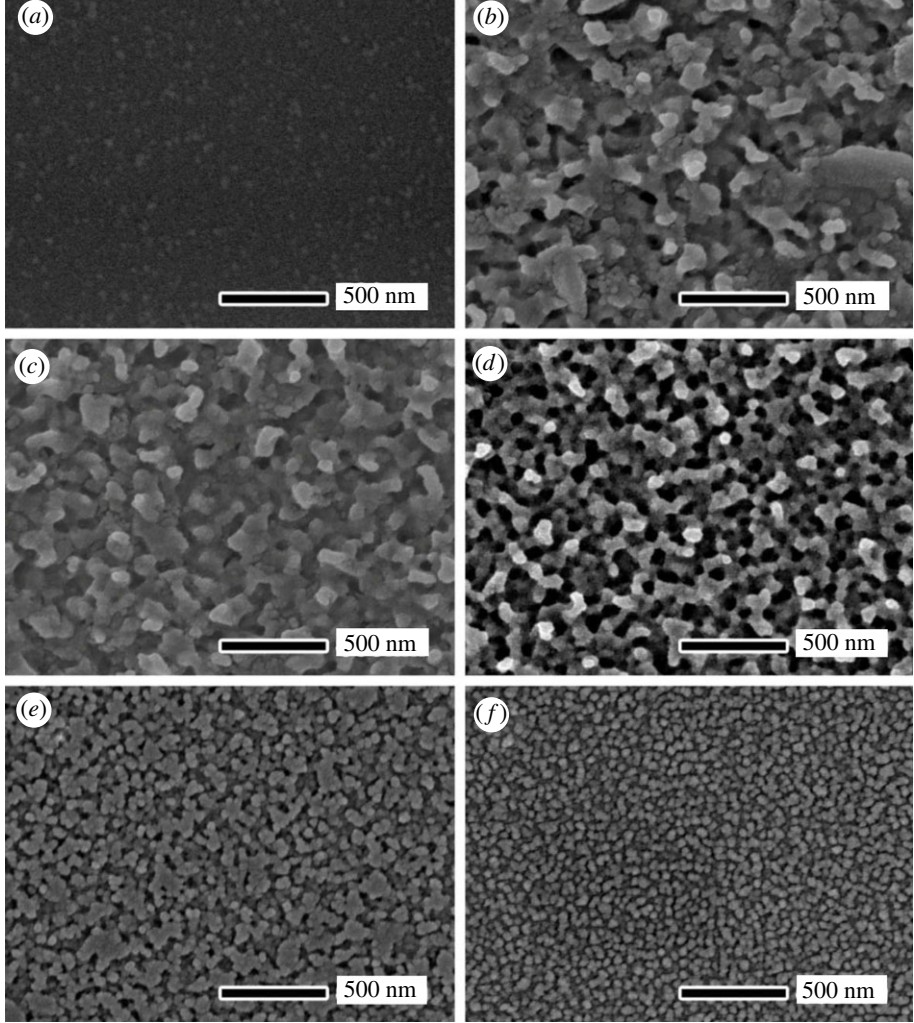

**Figure 5.** SEM images of the sample surface for different concentrations: (*a*) un-etched, (*b*) 1#, (*c*) 2#, (*d*) 3#, (*e*) 4# and (*f*) 5#.

**Table 1.** The concentrations of etching solutions.

| etching solutions | 1# | 2# | 3# | 4# | 5# |
|---|---|---|---|---|---|
| HF : $H_2O$ (volume ratio) | 1 : 1 | 1 : 2 | 1 : 4 | 1 : 9 | 1 : 19 |

excessive extension of the etching time (60 s), NPS was destroyed, and the pores were interconnected to form a channel with a width greater than 100 nm, which lost the anti-reflection effect. So, the optimum etching time was 50 s.

We also investigated the effect of etching solution concentrations on NPS morphology and performance. Table 1 shows the concentrations of different etching solutions. Figure 5 shows SEM images of the sample surface for different concentrations. It can be seen from figure 5 that the surface of the un-etched sample can achieve nano-level smoothness. The surface of etched sample became rough. After being etched by 1# solution with high concentration of hydrofluoric acid, the surface of the sample was greatly undulated, and the pores on the surface were very large and uneven. As the concentration of hydrofluoric acid decreased, the content of hydrofluoric acid in the droplets also gradually decreased. The degree of etched damage on the sample surface gradually decreased. After being etched by 3# etching solution, we can obtain dense, uniform and porous surface with pore diameter of about 50 nm. As the concentration of hydrofluoric acid was further reduced, the degree of corrosion was further reduced, and the diameter of the pores also gradually became smaller. Finally, we obtained a surface densely covered with uniform small protrusions (particle size about 10 nm) (5# etching solution).

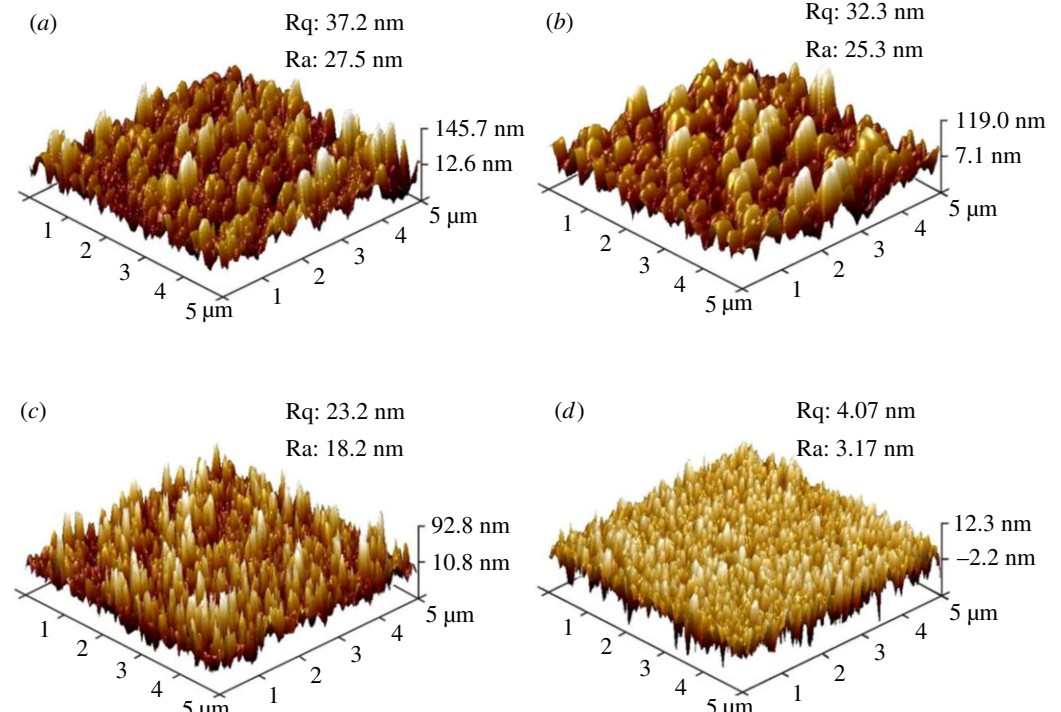

**Figure 6.** AFM images of the sample surface for different concentrations: (*a*) 1#, (*b*) 2#, (*c*) 3# and (*d*) 4#. Ra, arithmetic mean roughness; Rq, root mean square roughness.

Figure 6 shows AFM images of the sample surface for different concentrations. As the concentration of hydrofluoric acid decreased, the surface roughness (Ra) of the sample gradually decreased from 27.5 nm of the 1# etching solution to 3.17 nm of the 4# etching solution. Higher roughness was advantageous for the hydrophilicity. In order to obtain better hydrophilicity, the concentration of hydrofluoric acid should not be too low. The AFM diagram can also show the fluctuation of the sample surface from a three-dimensional perspective. Peak-like structure with narrow top and wide bottom was formed, and this structure was beneficial to the anti-reflection of the sample. In addition, by comparing the height scales of each sample, it was found that the height of the porous structure gradually decreased as the concentration of hydrofluoric acid decreased. The reason for this phenomenon was mainly that the etching solution will have a certain degree of volatile loss during the process of atomization and transportation, but the content of hydrofluoric acid in its droplets was the same as the concentration of hydrofluoric acid of the original etching solution, and the diameter and depth of the etched pits were positively related to the amount of hydrofluoric acid in the small droplets of acid fog. Therefore, the concentration of hydrofluoric acid in the etching solution directly affected the change of the sample surface structure.

Figure 7 shows transmittance spectra of etched samples for different concentrations. The average transmittance of samples in the visible light wavelength range (400–800 nm) was 90.9% (un-etched), 90.0% (1#), 91.9% (2#), 92.9% (3#) and 91.0% (4#). Obviously, as the concentration of hydrofluoric acid decreased, the average transmittance of the sample in the visible wavelength range increased first and then decreased. When the concentration of hydrofluoric acid was too high (1#), the surface of the sample was excessively etched, the diameter of the pores was large, the depth was deep and the distribution was extremely uneven, which was detrimental to the anti-reflection, so the average transmittance in the visible light wavelength range was lower than that of the un-etched sample. Then, the concentration of hydrofluoric acid was reduced, the distribution of the pores on the sample surface was gradually uniform, the diameter gradually became smaller, and the depth became shallower. The sample etched by solution 3# achieved the best anti-reflection effect. Its average transmittance was 2% higher than the original sample. If the concentration of the etching solution was too low, the etched degree of the sample surface became slight, the diameter of the pores became small and the depth was shallow. This structure resulted in low anti-reflection effect (4#).

Therefore, in order to obtain the best anti-reflection effect, the 3# etching solution, that is, the volume ratio of hydrofluoric acid to water is 1 : 4, was used as the optimal concentration.

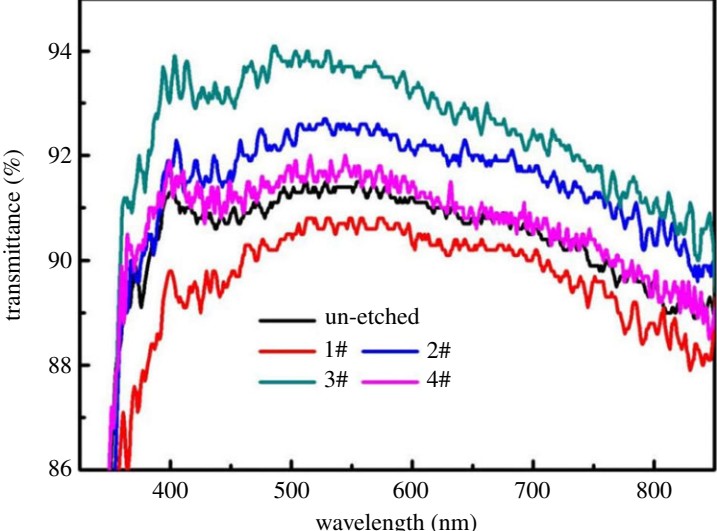

**Figure 7.** Transmittance spectra of etched samples for different concentrations.

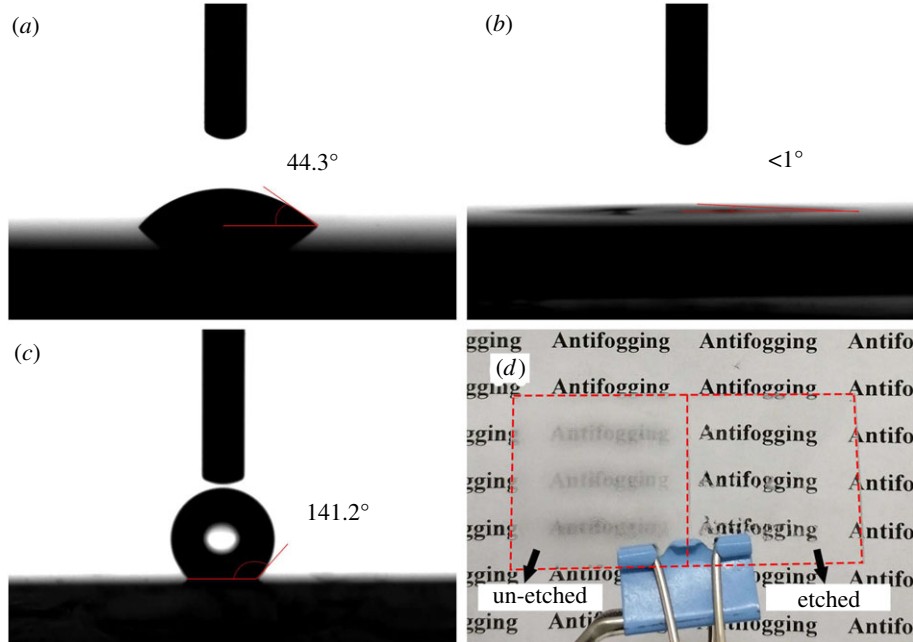

**Figure 8.** The water contact angle measurement for the un-etched sample (*a*), the etched sample (*b*), the fluorosilane passivated sample (*c*) and the anti-fog photograph (*d*).

Generally, the nanostructure surface tends to form the superhydrophilic or superhydrophobic property due to the large specific surface area [18]. The fog etching method can easily form NPS that is expected to have superior infiltration properties. Figure 3*c*,*d* shows the AFM images and the surface roughness of the original sample and the 50 s etching sample, respectively. Obviously, the original sample kept nano-scale smooth with the surface roughness Ra of 0.4 nm. The etched sample showed nano-scale roughness with Ra of 18.2 nm. For hydrophilic surfaces with a contact angle of less than 90°, the coarse structure made the hydrophilic surface more hydrophilic, while for hydrophobic surfaces with a contact angle greater than 90°, the coarse structure made the hydrophobic surface more hydrophobic. Figure 8 shows the measured water contact angles of the un-etched sample, the etched sample and the etched sample with fluorosilane passivation treatment. It can be found that the un-etched sample showed the normal hydrophilia where a water droplet was measured to have a contact angle of 44.3°. After 50 s etching at 323 K, the sample showed the superhydrophilic property with a contact angle less than 1° after the hydrophilicity test. The anti-fog measurement was applied

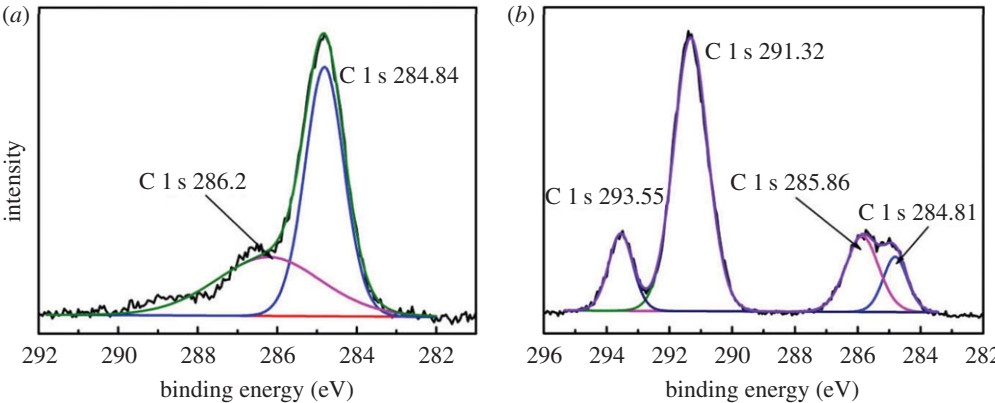

**Figure 9.** The XPS spectrum of C 1 s from the surface of the sample. (*a*) Original sample and (*b*) fluorosilane passivated sample.

at an etched sample which was cooled to 268 K and then exposed to the room temperature and atmospheric environment. Figure 8*d* shows the photograph of a glass which was etched at the right part. The written words located at the bottom of the glass can be clearly distinguished through the etched part glass. By contrast, the words were fuzzy at the bottom of the un-etched part glass. The fluorosilane passivated porous glass showed nearly the superhydrophobic property where a water droplet was measured to have a contact angle of 141.2°. Figure 9 is the XPS spectrum of C 1 s from the sample surface. Compared with figure 9*a*, figure 9*b* had two more peaks. In figure 9*b*, the peaks of 293.55 and 291.32 eV were representative of $CF_3$ and $CF_2$, respectively. $CF_3$ and $CF_2$ were both hydrophobic groups, the glass treated by acid fog was so rough that the fluorosilane passivated sample was nearly superhydrophobic.

## 4. Conclusion

A uniform porous layer on glass was prepared by a simple, inexpensive, efficient acid fog etching method. Different morphologies and structures of nanoporous layer can be obtained by etching at different time. The enhanced average transmittance of 92.9% is obtained over the whole visible light spectrum compared with that of 90.9% for an original sample. Especially, protosomatic NPS shows the superhydrophilic property with a contact angle less than 1° and presents an excellent anti-fog property. After fluorosilane passivation, NPS shows nearly the superhydrophobic property with a contact angle of 141.2°. This is a very effective method for preparing superhydrophilic, anti-fog, high-transmittance glass and is expected to achieve large-scale industrial production.

Data accessibility. All experimental materials and data can be accessed from the Dryad Digital Repository: https://doi.org/10.5061/dryad.j6q573n8n [19].
Authors' contributions. L.W. was responsible for proposing ideas and writing manuscript. L.L. and Y.L. designed and executed the experiment. S.W., H.C., H.J., Q.T. and W.S. measured the data, and D.Y. retouched the paper. All authors gave final approval for publication.
Competing interests. We declare that we have no competing interests.
Funding. This work was supported by the National Key R&D Program of China (grant no. 2018YFB1500300); National Natural Science Foundation of China (grant nos. 51532007, 61721005) and Key Project of Zhejiang Province (grant no. 2018C01034).
Acknowledgements. We thank Yali Ruan and MeiHuiZi Wang for their assistance with running participants for this experiment.

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
