## [Reviewer comments · Royal Society Open Science]

Review History

RSOS-192029.R0 (Original submission)

Review form: Reviewer 1

Is the manuscript scientifically sound in its present form?

No

Are the interpretations and conclusions justified by the results?

No

Is the language acceptable?

Yes

Do you have any ethical concerns with this paper?

No

Have you any concerns about statistical analyses in this paper?

No

Recommendation?

Major revision is needed (please make suggestions in comments)

Comments to the Author(s)

The novelty is not clear. HF vapor etching for glass porosification (and for other applications) is a well known method. The hydrophilic properties of the nanostructured surfaces and the modification of these properties using silanes are known as well.

The uniformity and reproducibility are not convincingly demonstrated.

The temperature is not indicated.

Optical simulation to see how the pore geometry influences the refractive index variation and the spectral reflectivity would be useful.

Review form: Reviewer 2

Is the manuscript scientifically sound in its present form?

Yes

Are the interpretations and conclusions justified by the results?

Yes

Is the language acceptable?

Yes

Do you have any ethical concerns with this paper?

No

Have you any concerns about statistical analyses in this paper?

No

Recommendation?

Accept with minor revision (please list in comments)

Comments to the Author(s)

The authors mainly fabricated a uniform porous layer on glass by a simple, inexpensive, efficient acid fog etching method. This idea seems practical in materials design, however, some errors can be found in the paper, and several detailed information are suggested to be discussed further, as presented below:

Overall problems:

1. Please elaborate on the significance of this topic, such as what function the material's superhydrophilicity can achieve. Where is superhydrophilic suitable, and where is superhydrophobic suitable?
2. Optimization of other technological parameters affecting the morphology and properties of NPS should be further discussed.
3. It is known that superhydrophilic surfaces generally do not retain their properties for very long. Experiments evaluating how long superhydrophilicity can be maintained need to be supplemented.
4. The pore conditions at different etching times were compared, but the optimum etching time was not indicated. Besides, although the ligament increased in 60s while the diameter of porosity decreased, evidences were not adequate to prove that prolonging etching time would result in enlarged pores. Hence more SEM images and conditions of etching time need to be provided to improve the veracity of your data. In addition, could the authors explain why the transmittance spectra of the nanoporous structures etching for 60s is lower than un-etched one, as

it is worth noticing that the etched samples whose etching time below 60s were all higher than un-etched one.

Details and errors:

1. In Figure.5, the bottom text “ample” should be replaced by “sample”.

Review form: Reviewer 3

Is the manuscript scientifically sound in its present form?

Yes

Are the interpretations and conclusions justified by the results?

Yes

Is the language acceptable?

No

Do you have any ethical concerns with this paper?

No

Have you any concerns about statistical analyses in this paper?

No

Recommendation?

Accept with minor revision (please list in comments)

Comments to the Author(s)

This manuscript describes a facile route to generate nanoporous structure (NPS) on glass surface using hydrofluoric acid fog formed by air-jet atomization. The results presented in the manuscript show that light transmittance as high as ~92.9% can be achieved on the treated glass. In addition, the hydrofluoric acid fog treated sample gives a water contact angle of less than 1°, comparable to most of the superhydrophilic glasses reported in the literature. The authors should address the following points before the manuscript is ready to be considered for publication:

In Section 1, the authors state that “the method is simple, cheap and high efficiency, which makes it very convenient to produce on a large scale”. Perhaps the authors should further address on the safety concern of using hydrofluoric acid in the industry.

Glass surface etching by hydrofluoric acid fog formed using air-jet atomization is one of the crucial steps that distinguish novelty of this study from others. More information on such an important step should be given. Was the atomizer home fabricated or commercially available? The authors should give the schematic representation of the experiment setup.

In Section 2, the authors state that “At the same temperature, the hydrofluoric acid (40 wt%, AR) fog formed by air-jet atomization...”; but in Section 3, the authors state that “...and 9.6 wt% hydrofluoric acid solution was used”. So what was the acid concentration used?

What are the water contact angles of etched samples prepared with different etching time? What is the water contact angle of the un-etched glass but with fluorosilane passivation?

There are numerous grammatical and typo errors throughout the manuscript that should be rectified.

For examples, (i) inconsistent tenses are used in Section 2 (lines 8-34) and some of the sentences are in active voices; (ii) "transmittances spectra", "hydrophily", "angel"

Decision letter (RSOS-192029.R0)

30-Mar-2020

Dear Mr Li:

Title: The preparation and characterization of uniform nanoporous structure on glass
Manuscript ID: RSOS-192029

The editor assigned to your manuscript has now received comments from reviewers. I apologise that this has taken longer than usual. We would like you to revise your paper in accordance with the referee and Subject Editor suggestions which can be found below (not including confidential reports to the Editor). Please note this decision does not guarantee eventual acceptance.

Please submit your revised paper before 22-Apr-2020. Please note that the revision deadline will expire at 00.00am on this date. If we do not hear from you within this time then it will be assumed that the paper has been withdrawn. In exceptional circumstances, extensions may be possible if agreed with the Editorial Office in advance. We do not allow multiple rounds of revision so we urge you to make every effort to fully address all of the comments at this stage. If deemed necessary by the Editors, your manuscript will be sent back to one or more of the original reviewers for assessment. If the original reviewers are not available we may invite new reviewers.

RSC Associate Editor:
Comments to the Author:
(There are no comments.)

RSC Subject Editor:
Comments to the Author:
(There are no comments.)

Reviewers' Comments to Author:
Reviewer: 1

Comments to the Author(s)
The novelty is not clear. HF vapor etching for glass porosification (and for other applications) is a well known method. The hydrophylic properties of the nanostructured surfaces and the modification of these properties using silanes are known as well.

The uniformity and reproducibility are not convincing demonstrated.
The temperature is not indicated.
Optical simulation to see how the pores geometry influence the refractive index variation and the spectral reflectivity would be useful.

Reviewer: 2

Comments to the Author(s)
The authors mainly fabricated a uniform porous layer on glass by a simple, inexpensive, efficient acid fog etching method. This idea seems practical in materials design, however, some errors can be found in the paper, and several detailed information are suggested to be discussed further, as presented below:

Overall problems:

1. Please elaborate on the significance of this topic, such as what function the material's superhydrophilicity can achieve. Where is superhydrophilic suitable, and where is superhydrophobic suitable?
2. Optimization of other technological parameters affecting the morphology and properties of NPS should be further discussed.
3. It is known that superhydrophilic surfaces generally do not retain their properties for very long. Experiments evaluating how long superhydrophilicity can be maintained need to be supplemented.
4. The pore conditions at different etching times were compared, but the optimum etching time was not indicated. Besides, although the ligament increased in 60s while the diameter of porosity decreased, evidences were not adequate to prove that prolonging etching time would result in enlarged pores. Hence more SEM images and conditions of etching time need to be provided to improve the veracity of your data. In addition, could the authors explain why the transmittance spectra of the nanoporous structures etching for 60s is lower than un-etched one, as it is worth noticing that the etched samples whose etching time below 60s were all higher than un-etched one.

Details and errors:

1. In Figure.5, the bottom text “ample” should be replaced by “sample”.

Reviewer: 3

Comments to the Author(s)

This manuscript describes a facile route to generate nanoporous structure (NPS) on glass surface using hydrofluoric acid fog formed by air-jet atomization. The results presented in the manuscript show that light transmittance as high as ~92.9% can be achieved on the treated glass. In addition, the hydrofluoric acid fog treated sample gives a water contact angle of less than 1°, comparable to most of the superhydrophilic glasses reported in the literature. The authors should address the following points before the manuscript is ready to be considered for publication:

In Section 1, the authors state that “the method is simple, cheap and high efficiency, which makes it very convenient to produce on a large scale”. Perhaps the authors should further address on the safety concern of using hydrofluoric acid in the industry.

Glass surface etching by hydrofluoric acid fog formed using air-jet atomization is one of the crucial steps that distinguish novelty of this study from others. More information on such an important step should be given. Was the atomizer home fabricated or commercially available? The authors should give the schematic representation of the experiment setup.

In Section 2, the authors state that “At the same temperature, the hydrofluoric acid (40 wt%, AR) fog formed by air-jet atomization...”; but in Section 3, the authors state that “...and 9.6 wt% hydrofluoric acid solution was used”. So what was the acid concentration used?

What are the water contact angles of etched samples prepared with different etching time? What is the water contact angle of the un-etched glass but with fluorosilane passivation?

There are numerous grammatical and typo errors throughout the manuscript that should be rectified.

For examples, (i) inconsistent tenses are used in Section 2 (lines 8-34) and some of the sentences are in active voices; (ii) “transmittances spectra”, “hydrophily”, “angel”

Author's Response to Decision Letter for (RSOS-192029.R0)

See Appendix A.

RSOS-192029.R1 (Revision)

Review form: Reviewer 3

Is the manuscript scientifically sound in its present form?

No

Are the interpretations and conclusions justified by the results?

Yes

Is the language acceptable?

Yes

Do you have any ethical concerns with this paper?

No

Have you any concerns about statistical analyses in this paper?

No

Recommendation?

Accept with minor revision (please list in comments)

Comments to the Author(s)

The authors have addressed most of the concerns raised by the previous reviewers. I recommend that this manuscript to be accepted for publication after minor revisions.

The main contents from the following should be included in the revised manuscript:

- (1) Response to Comment 3 of Reviewer 1
- (2) Response to Comment 1 of Reviewer 2, with relevant reference(s) cited
- (3) Response to Comment 2 of Reviewer 2
- (4) Response to Comment 4 of Reviewer 2

Decision letter (RSOS-192029.R1)

Dear Mr Li:

Title: The preparation and characterization of uniform nanoporous structure on glass
Manuscript ID: RSOS-192029.R1

Thank you for submitting the above manuscript to Royal Society Open Science. On behalf of the Editors and the Royal Society of Chemistry, I am pleased to inform you that your manuscript will be accepted for publication in Royal Society Open Science subject to minor revision in accordance with the referee suggestions. Please find the reviewers' comments at the end of this email.

The reviewers and handling editors have recommended publication, but also suggest some minor revisions to your manuscript. Therefore, I invite you to respond to the comments and revise your manuscript.

Because the schedule for publication is very tight, it is a condition of publication that you submit the revised version of your manuscript before 21-May-2020. Please note that the revision deadline will expire at 00.00am on this date. If you do not think you will be able to meet this date please let me know immediately.

Kind regards,
Dr Laura Smith
Publishing Editor, Journals

RSC Associate Editor:
Comments to the Author:
(There are no comments.)

RSC Subject Editor:
Comments to the Author:
(There are no comments.)

Reviewer comments to Author:
Reviewer: 3

Comments to the Author(s)
The authors have addressed most of the concerns raised by the previous reviewers. I recommend that this manuscript to be accepted for publication after minor revisions.

The main contents from the following should be included in the revised manuscript:

- (1) Response to Comment 3 of Reviewer 1
- (2) Response to Comment 1 of Reviewer 2, with relevant reference(s) cited
- (3) Response to Comment 2 of Reviewer 2
- (4) Response to Comment 4 of Reviewer 2

Author's Response to Decision Letter for (RSOS-192029.R1)

See Appendix B.

RSOS-192029.R2 (Revision)

Review form: Reviewer 3

Is the manuscript scientifically sound in its present form?

Yes

Are the interpretations and conclusions justified by the results?

Yes

Is the language acceptable?

Yes

Do you have any ethical concerns with this paper?

No

Have you any concerns about statistical analyses in this paper?

No

Recommendation?

Accept as is

Comments to the Author(s)

The authors have addressed all of the concerns raised by the previous reviewer. I recommend that this manuscript to be accepted for publication.

Decision letter (RSOS-192029.R2)

Dear Mr Li:

Title: The preparation and characterization of uniform nanoporous structure on glass
Manuscript ID: RSOS-192029.R2

It is a pleasure to accept your manuscript in its current form for publication in Royal Society Open Science. The chemistry content of Royal Society Open Science is published in collaboration with the Royal Society of Chemistry.

RSC Associate Editor:
Comments to the Author:
(There are no comments.)

RSC Subject Editor:
Comments to the Author:
(There are no comments.)

Reviewer(s)' Comments to Author:
Reviewer: 3

Comments to the Author(s)
The authors have addressed all of the concerns raised by the previous reviewer. I recommend that this manuscript to be accepted for publication.

Lei Wang, Ph.D.

Associate Professor for Semiconductor Materials

State Key Laboratory of Silicon Materials

Department of Materials Science and Engineering

Dear reviewers,

Thank you very much for your constructive comments and valuable suggestions on our manuscript. We have revised the manuscript according to your suggestions. In the following, point-by-point responses to your comments/suggestions are detailed. In the manuscript, I have marked all changes in red font.

Reviewer 1

Comment 1. The uniformity and reproducibility are not convincing demonstrated.

Response:

We repeated the experiment dozens of times, and the uniformity and repeatability of the samples are reliable.

Comment 2. The temperature is not indicated.

Response:

In the experiment, different etching temperatures are 303 K, 323 K, 328 K, 333 K, respectively. The optimal etching temperature is 323 K.

Comment 3. Optical simulation to see how the pores geometry influence the refractive index variation and the spectral reflectivity would be useful.

Response:

In the early stage of the etching reaction (20-30 s), the sample was gradually etched, and the surface began to produce peak-like structure. As the reaction progressed (40-50 s), the distribution of this peak-like structure became uniform and dense, According to effective medium theory:

$$n_1 = [n_g^2 f + n_{air}^2 (1 - f)]^{1/2}$$

n_1 , n_g and n_{air} are the refractive index of NPS, glass and air, respectively. f is the fill factor, which is the percentage of the peak structure in the porous structure.

According to the analysis of the incident path of visible light, at the outermost surface of the sample, $f = 0$, $n_1 = n_{\text{air}}$. Down along the peak-like structure, the structure narrowed first and then widened, f increased gradually from 0 to 1, n_1 increased from n_{air} to n_g . It can be considered that NPS as the AR coating had a graded refractive index, so that transmittance can enhance. However, with the excessive extension of the etching time (60 s), this peak-like structure was destroyed, and the pores were interconnected to form a channel with a width greater than 100 nm, which lost the antireflection effect.

Reviewer 2

Comment 1. Please elaborate on the significance of this topic, such as what function the material's superhydrophilicity can achieve. Where is superhydrophilic suitable, and where is superhydrophobic suitable?

Response:

Superhydrophilic or superhydrophobic glasses with high transmittance can be used in the fields of optics, architecture, energy, etc. The super-hydrophilic glasses have the spreading performance of water droplets, which can prevent the diffuse reflection of water droplets and reduce the static electricity on the surface, which can be used for anti-fog and suppression of small particles adsorption. For example, they can be used in myopia glasses, endoscopes, car windshields and other occasions. The super-hydrophobic glasses can make the water droplets slide down quickly, inhibit the adsorption of large particles and sewage on the surface, and can be used in automobile side mirrors, solar cell module cover plates and other occasions. Generally, super-hydrophilic and super-hydrophobic glasses can achieve self-cleaning of the glass surface and maintain high light transmittance.

Comment 2. Optimization of other technological parameters affecting the morphology and properties of NPS should be further discussed.

Response:

We also investigated the effect of etching solution concentrations on NPS morphology and

performance. Table 1 shows the concentrations of different etching solutions. Fig. 1 shows SEM images of the sample surface for different concentrations. It can be seen from Fig. 1 that the surface of the un-etched sample can achieve nano-level smoothness. The surface of etched sample became rough. After being etched by 1# solution with high concentration of hydrofluoric acid, the surface of the sample was greatly undulated, and the pores on the surface were very large and uneven. As the concentration of hydrofluoric acid decreased, the content of hydrofluoric acid in the droplets also gradually decreased. The degree of etched damage on the sample surface gradually decreased. After being etched by 3# etching solution, we can obtain dense, uniform and porous surface with pore diameter of about 50 nm. As the concentration of hydrofluoric acid was further reduced, the degree of corrosion was further reduced, and the diameter of the pores also gradually became smaller. Finally, we obtained a surface densely covered with uniform small protrusions (particle size about 10 nm) (5# corrosive liquid).

Table 1 The concentrations of etching solutions

etching solutions	1#	2#	3#	4#	5#
HF : H ₂ O (volume ratio)	1:1	1:2	1:4	1:9	1:19

Fig. 1. SEM images of the sample surface for different concentrations: (a) un-etched;(b) 1#;(c) 2#;(d) 3#;(e) 4#;(f) 5#

Fig. 2. AFM images of the sample surface for different concentrations:(a) 1#;(b) 2#;(c) 3#;(d) 4#

Fig. 2 shows AFM images of the sample surface for different concentrations. As the concentration of hydrofluoric acid decreased, the surface roughness (Ra) of the sample gradually decreased from 27.5 nm of the 1# etching solution to 3.17 nm of the 4# etching solution. Higher roughness was advantageous for the hydrophilicity. In order to obtain better hydrophilicity, the concentration of hydrofluoric acid should not be too low. The AFM diagram can also show the fluctuation of the sample surface from a three-dimensional perspective. Peak-like structure with narrow top and wide bottom was formed, and this structure was beneficial to the anti-reflection of the sample. In addition, by comparing the height scales of each sample, it was found that the height of the porous structure gradually decreased as the concentration of hydrofluoric acid decreased. The reason for this phenomenon was mainly that the etching solution will have a certain degree of volatile loss during the process of atomization and transportation, but the content of hydrofluoric acid in its droplets was the same as the concentration of hydrofluoric acid of the original etching solution, and the diameter and depth of the etched pits were positively related to the amount of

hydrofluoric acid in the small droplets of acid fog. Therefore, the concentration of hydrofluoric acid in the etching solution directly affected the change of the sample surface structure.

Fig. 3. Transmittance spectra of etched samples for different concentrations

Fig. 3 shows transmittance spectra of etched samples for different concentrations. The average transmittance of samples in the visible light wavelength range (400-800 nm) were 90.9% (un-etched), 90.0% (1#), 91.9% (2#), 92.9% (3#) and 91.0% (4#). Obviously, as the concentration of hydrofluoric acid decreased, the average transmittance of the sample in the visible wavelength range increased first and then decreased. When the concentration of hydrofluoric acid was too high (1#), the surface of the sample was excessively etched, the diameter of the pores was large, the depth was deep and the distribution was extremely uneven, which was detrimental to the anti-reflection, so the average transmittance in the visible light wavelength range was lower than that of the un-etched sample. Then, the concentration of hydrofluoric acid was reduced, the distribution of the pores on the sample surface was gradually uniform, the diameter gradually became smaller, and the depth became shallower. The sample etched by solution 3# achieved the best anti-reflection effect. Its average transmittance was 2% higher than the original sample. If the concentration of the etching solution was too low, the etched degree of the sample surface became slight, the diameter of the pores became small and the depth was shallow. This structure resulted in low anti-

reflection effect (4#).

Therefore, in order to obtain the best anti-reflection effect, the 3# etching solution, that was, the volume ratio of hydrofluoric acid to water is 1:4, was used as the optimal concentration.

Comment 3. It is known that superhydrophilic surfaces generally do not retain their properties for very long. Experiments evaluating how long superhydrophilicity can be maintained need to be supplemented.

Response:

Sorry, the test has not yet been carried out systematically, and we will conduct research in this area in future work.

Comment 4. The pore conditions at different etching times were compared, but the optimum etching time was not indicated. Besides, although the ligament increased in 60s while the diameter of porosity decreased, evidences were not adequate to prove that prolonging etching time would result in enlarged pores. Hence more SEM images and conditions of etching time need to be provided to improve the veracity of your data. In addition, could the authors explain why the transmittance spectra of the nanoporous structures etching for 60s is lower than un-etched one, as it is worth noticing that the etched samples whose etching time below 60s were all higher than un-etched one.

Response:

The optimum etching time was 50 s.

In the early stage of the etching reaction (20-30 s), the sample was gradually etched, and the surface began to produce peak-like structure. As the reaction progressed (40-50 s), the distribution of this peak-like structure became uniform and dense, According to effective medium theory:

$$n_1 = [n_g^2 f + n_{air}^2 (1 - f)]^{1/2}$$

n_1 , n_g and n_{air} are the refractive index of NPS, glass and air, respectively. f is the fill factor, which is the percentage of the peak structure in the porous structure.

According to the analysis of the incident path of visible light, at the outermost surface of the sample, $f = 0$, $n_1 = n_{air}$. Down along the peak-like structure, the structure narrowed first and then widened, f increased gradually from 0 to 1, n_1 increased from n_{air} to n_g . It can be considered that

NPS as the AR coating had a graded refractive index, so that transmittance can enhance. However, with the excessive extension of the etching time (60 s), this peak-like structure was destroyed, and the pores were interconnected to form a channel with a width greater than 100 nm, resulting in the diffuse reflection of the light and low transmittance.

Reviewer 3

Comment 1. In Section 1, the authors state that “the method is simple, cheap and high efficiency, which makes it very convenient to produce on a large scale”. Perhaps the authors should further address on the safety concern of using hydrofluoric acid in the industry.

Response:

The safety of hydrofluoric acid is indeed a problem in industry. The experiment needs to be conducted in a completely enclosed space. If industrial production is required, it also needs to be carried out in a enclosed space.

Comment 2. Glass surface etching by hydrofluoric acid fog formed using air-jet atomization is one of the crucial steps that distinguish novelty of this study from others. More information on such an important step should be given. Was the atomizer home fabricated or commercially available? The authors should give the schematic representation of the experiment setup.

Response:

The model number of atomizer is Yu Yue 403T purchased from Yuwell-Jiangsu Yuyue medical equipment & supply Co., Ltd. . Fig. 4 shows the schematic representation of the experiment setup.

Fig. 4. The schematic representation of the experiment setup

Comment 3. In Section 2, the authors state that “At the same temperature, the hydrofluoric acid (40 wt%, AR) fog formed by air-jet atomization...”; but in Section 3, the authors state that “...and 9.6 wt% hydrofluoric acid solution was used”. So what was the acid concentration used?

Response:

In section 2, 40 wt% refers to the concentration of purchased hydrofluoric acid. In experiment, 9.6 wt% refers to the concentration of hydrofluoric acid in the etching solution.

Comment 4. What are the water contact angles of etched samples prepared with different etching time? What is the water contact angle of the un-etched glass but with fluorosilane passivation?

Response:

Sorry, the contact angle test has not yet been carried out systematically, and we will conduct research in this area in future work.

Comment 5. There are numerous grammatical and typo errors throughout the manuscript that should be rectified.

For examples, (i) inconsistent tenses are used in Section 2 (lines 8-34) and some of the sentences are in active voices; (ii) “transmittances spectra”, “hydrophily”, “angel”

Response:

We are so sorry for the grammatical and typo errors, and we try our best to correct these errors.

Hopefully, we have essentially addressed your concerns and comments. Thank you again for all your pertinent comments that enable us to refine our manuscript.

Yours sincerely,

Lei Wang

Appendix B

Lei Wang, Ph.D.

Associate Professor for Semiconductor Materials

State Key Laboratory of Silicon Materials

Department of Materials Science and Engineering

Dear reviewers,

Thank you very much for your constructive comments and valuable suggestions on our manuscript. We have revised the manuscript according to your suggestions. In the following, point-by-point responses to your comments/suggestions are detailed. In the manuscript, I have marked all changes in red font.

Reviewer 3

Comment. The main contents from the following should be included in the revised manuscript:

(1) Response to Comment 3 of Reviewer 1

(2) Response to Comment 1 of Reviewer 2, with relevant reference(s) cited

(3) Response to Comment 2 of Reviewer 2

(4) Response to Comment 4 of Reviewer 2

Response:

I have added the above contents to the draft manuscript. In the manuscript, I have marked all changes in red font.

Hopefully, we have essentially addressed your concerns and comments. Thank you again for all your pertinent comments that enable us to refine our manuscript.

Yours sincerely,

Lei Wang

Zheda Road 38#, Hangzhou 310027, China

phy_wangl@zju.edu.cn